# Methods for Radiolabeling Nanoparticles (Part 3): Therapeutic Use

**DOI:** 10.3390/biom13081241

**Published:** 2023-08-12

**Authors:** Valeria Bentivoglio, Pallavi Nayak, Michela Varani, Chiara Lauri, Alberto Signore

**Affiliations:** Nuclear Medicine Unit, Department of Medical-Surgical Sciences and of Translational Medicine, Faculty of Medicine and Psychology, “Sapienza” University of Rome, 00185 Rome, Italy; valeria.bentivoglio@uniroma1.it (V.B.); pallavi.nayak@uniroma1.it (P.N.); michela.varani@uniroma1.it (M.V.); chiara.lauri@uniroma1.it (C.L.)

**Keywords:** nanoparticles, nanotechnology, nuclear medicine, radiolabeling, cancer therapy

## Abstract

Following previously published systematic reviews on the diagnostic use of nanoparticles (NPs), in this manuscript, we report published methods for radiolabeling nanoparticles with therapeutic alpha-emitting, beta-emitting, or Auger’s electron-emitting isotopes. After analyzing 234 papers, we found that different methods were used with the same isotope and the same type of nanoparticle. The most common type of nanoparticles used are the PLGA and PAMAM nanoparticles, and the most commonly used therapeutic isotope is ^177^Lu. Regarding labeling methods, the direct encapsulation of the isotope resulted in the most reliable and reproducible technique. Radiolabeled nanoparticles show promising results in metastatic breast and lung cancer, although this field of research needs more clinical studies, mainly on the comparison of nanoparticles with chemotherapy.

## 1. Introduction

In the last ten years, the field of nanomedicine and associated research has enormously expanded, without a significant reduction due to the COVID pandemic in the previous three years.

This exponential growth in publications is particularly evident if we use “nanoparticles” as a search input, rising from 17,046 in 2012 to 34,132 in 2022 (Figure 1). 

A few of these publications discuss the use of “radiolabeled nanoparticles”, which include “radiolabeled nanoparticles for diagnostic SPECT use”, “radiolabeled nanoparticles for diagnostic PET use”, and “radiolabeled nanoparticles for therapy” (Figure 1).

In general, the methods for NP measurement in biological samples depend on the chemical composition and structure of NPs, and there is no universal technique that is suitable for all NPs. Among the techniques established for NP measurement and imaging, isotope tracing, particularly radioactive tracing, is one of the most powerful for assigning a source and monitoring its distribution from the nano to the global scale. Furthermore, radiolabeled NPs have received significant attention in nuclear medicine, molecular imaging, drug delivery, and radiation therapy [1,2]. 

We previously published two systematic reviews on “methods for radiolabeling nanoparticles for diagnostic SPECT use” [3] and “methods for radiolabeling nanoparticles for diagnostic PET use” [4]. This paper addresses a systematic review of the published research on “methods for radiolabeling nanoparticles for therapy”, including an Auger’s electron-emitting, Alpha-emitting, and Beta-emitting isotopes. 

## 2. Radiolabeling of NPs with Auger’s Electron-Emitting Isotopes

Auger’s electrons are emitted from radioactive nuclei due to the Auger effect. The Auger effect is the emission of an Auger’s electron from an atom that has gained energy by filling an external electronic shell, which fills a vacancy in an external electronic orbital that can occur during radioactive decay.

Auger’s electrons have a high level of linear energy transfer (LET) (1–26 keV mm^−1^) and a very small particle range (<0.5 mm). These properties enable the radioisotope to be transported intracellularly to the nucleus, maximizing cytotoxic activity, which results in DNA double-strand breaks [5].

Several isotopes can emit Auger’s electrons, such as ^64^Cu, ^161^Tb, ^195m^Pt, ^111^In, ^67^Ga, ^99m^Tc, ^125^I, ^124^I, and ^123^I. However, most of these isotopes are used for diagnostic purposes and have already been described in the Part 1 and Part 2 reviews [3,4].

Here, we describe the labeling methods using ^64^Cu, one of the most commonly used radioisotopes for therapeutic purposes, as summarized in Table 1.

### 2.1. Radiolabeling with Copper-64

Copper-64 (^64^Cu) is a cyclotron-produced radionuclide with a half-life of 12.7 h, and it decays through β^+^ emission (17%), β^−^ emission (39%), and Auger’s electrons (44%). These features make ^64^Cu a suitable radioisotope for radiotherapy by β^−^ particles and Auger’s electrons and for radio imaging, including positron-emission tomography (PET) imaging by β^+^ particles, in cancer. The decaying of ^64^Cu makes it a perfect radionuclide for theragnostics [6,16].

#### 2.1.1. Direct Radiolabeling

The ^64^Cu has been used to directly and indirectly label different NPs. Copper-sulfide nanoparticles (Cu_2_S NPs) are among the NPs that can be labeled by the direct method and show therapeutic effects. The synthesis of these NPs begins with the introduction of Na_2_S into an aqueous solution of CuCl_2_ together with an aliquot of ^64^CuCl_2_, in the presence of PEG-SH. The rapidity of this method, which takes only 20 min, is important for minimizing radioactive decay. The radiolabeling efficiency and stability were analyzed using instant thin-layer chromatography (iTLC), and the results showed that more than 99% of the radioactivity was associated with the radiolabeled NPs at the end of synthesis [6].

Gailkwad et al. proposed another strategy for direct radiolabeling with NPs. Chitosan (CHS) was first radiolabeled and then converted to NPs through ionotropic gelation. The authors tried this method with several therapeutic radioisotopes, such as ^64^Cu, ^90^Y, ^166^Ho, and ^177^Lu. The radiolabeling process started with the mixing of the CHS solution with a water-soluble chitosan (COS) solution, and, after adjusting the pH to 5 using NaOH, the solution was stirred for 30 min at room temperature. Subsequently, the relevant radiometal solution was added with TPP and stirred for 10 min, producing radioactive-CH NPs. The chitosan NPs were cross-linked with a glutaraldehyde solution while stirring for at least 30 min. The radiochemical purity was determined using iTLC and was greater than 98% for all the samples. This novel technique takes advantage of chitosan’s unique physical and chemical properties to provide an easy, rapid, and highly selective radiolabeling strategy [7].

Instead of chitosan, other organic materials can also be used for the synthesis of radiolabeled nanoparticles; Zhou et al. fabricated radiolabeled melanin NPs (MNPs) with a chelator-free approach, mixed them with ^64^CuCl_2_ in the presence of a sodium acetate (NaOAc) buffer, and left the mixture to incubate for 1 h at 40 °C. An Al aliquot of trifluoroacetic acid (TFA) was used to stop the reaction.

The radiochemical purity (RCP) was evaluated by ITLC, and it was found to be greater than 95%. The therapeutic effect was evaluated on human-epidermoid-carcinoma-tumor-bearing nude mice, and the findings showed a positive result, in that the tumor growth was significantly inhibited after treatment with a single dose (55.5 MBq). The comprehensive study concluded that ^64^Cu-MNPs is a promising candidate for cancer therapy [8].

Gholami et al. developed radiolabeled NPs with superparamagnetic iron oxide (SPIO) cores, commonly known as Feraheme^®^ (FH) NPs, in which radioisotopes bind directly to the surface of the SPIO core. The therapeutic radioisotopes used were ^223^Ra, ^177^Lu, ^90^Y, and ^64^Cu. The study showed that the FH NPs could be radiolabeled with high RCY and RCP using both diagnostic radioisotopes with long-range beta emitters and therapeutic radioisotopes with short-range alpha emitters [9].

Boron nitride nanotubes (BNNTs) are cylindrical structures formed by atoms of boron (B) and nitrogen (N). The empty internal spaces allow BNNTs to occupy various chemical species, including radioisotopes, in their vacant sites. The solvothermal process is mainly used to radiolabel BNNTs with ^64^Cu. In this procedure, the BNNTs were dispersed in anhydrous ethanol, and the ^64^Cu was encapsulated using an autoclave with a polytetrafluoroethylene (PTFE) vessel for 2 h at 180 °C. This method resulted in a stable and contaminant-free radioisotope, and the TEM images showed that the BNNTs were structurally well organized, presenting ^64^Cu nanoparticles not only in their internal channels but also on their surfaces [10].

All of these studies convey the fact that radiolabeling occurs during nanoparticle synthesis. However, Xie et al. studied a post-synthesis approach that allows the encapsulation of ^64^Cu into NPs, particularly covellite nanocrystals (CuS NCs). The researchers demonstrated that the covellite NCs dispersed with toluene incorporated Cu(I) ions [11]. Consequently, the in situ reduction of ^64^Cu (II) to ^64^Cu(I) was required. For this purpose, ascorbic acid, a nontoxic and mild reducing agent, was added to the solution. The reactions on the NCs were tested in an aqueous environment, and most of the incorporation process was complete within a few minutes at room temperature [12].

A new carrier system based on a simple single-step precipitation technique was designed to synthesize radioactive barium sulfate nanoparticles (BaSO_4_ NPs) with excellent stability in an aqueous environment (<5% activity release). In this work, the incorporation of several radionuclides, such as ^131^Ba, ^133^Ba, ^224^Ra, ^18^F, ^64^Cu, ^89^Zr, ^111^In, and ^177^Lu, was investigated. The synthesis and radiolabeling process forecasts barium-alendronate precipitation by adding ethanol and the respective radionuclide, which provide nano-templates for the subsequent growth of the alendronate-containing BaSO_4_ NPs. The radio-chemical yield (RCY) was expressed as the ratio of the amount of activity in the NP pellets to the starting activity, and while the ^177^Lu- BaSO_4_ NPs had an acceptable RCY, of 69%, the ^64^Cu-BaSO_4_ NPs did not achieve sufficient labeling due to the affinity of the alendronate as a complexing agent for copper ions [13].

The neutron irradiation of the non-radioactive isotopes that comprise NPs is another method for obtaining radioactive NPs. The hydrothermal coprecipitation method was used to synthesize hydroxyapatite (HA) NPs with copper. The synthesis started with the introduction of two precursor solutions: the first was prepared by mixing in water cetyl-trimethyl-ammonium-bromide (CTAB), calcium nitrate tetrahydrate (Ca(NO_3_)_2_·4H_2_O,) and copper(II) nitrate hydrate (Cu(NO_3_)_2_·3H_2_O). In contrast, the second solution was obtained by mixing potassium phosphate dibasic trihydrate (K_2_HPO_4_·3H_2_O) with water. After adjusting the pH of both solutions to 12, the precursor solution was left for 6 h, and then the second solution was added dropwise to the first solution. The final product was stirred continuously throughout the night before being transferred to an autoclave to continue the hydrothermal treatment at 100 °C for 10 h. The materials were calcined at 600 °C under air pressure for 6 h after centrifugation and vacuum filtering to remove the organic template and grow the tenorite phase. The NPs were irradiated for 2 h with a thermal neutron flow to activate the ^64^Cu [14].

#### 2.1.2. Indirect Radiolabeling

Most of the previous studies involved the use of ^64^Cu for a diagnostic purpose, which we discussed in our previous review [3]. However, there are few works in the literature on the indirect labeling of nanoparticles with ^64^Cu for therapeutic purposes.

Rossin et al. studied folate-conjugated shell-cross-linked nanoparticles (SCKs) functionalized with 1,4,8,11-tetraazacyclotetradecane-N, N′, N″, and N‴-tetraacetic acid (TETA). This bifunctional chelator (BFC) is a macrocyclic ligand that provides thermodynamic stability and copper-complex kinetics in vivo. To obtain these NPs, a multi-step process was required. The authors proceeded to synthesize the BFC after conjugating the SCK with folate, allowing them to react with a mono tert-Boc-protected diamine (N-butoxycarbonyl-[2,2′-(ethylenedioxy) bis(ethylamine)]) with TETA in the presence of EDC for 16 h. The tert-Boc-protecting group was cleaved using trifluoroacetic acid, and the TETA–NH_2_ product was purified via methanol and acetone precipitation. The carboxylic groups on the SCK–folate shells were activated by the EDC-NHs system in a PBS solution for 2 h at 4 °C while stirring. The TETA-NH2 was added after the excess EDC-NHS was separated, and the reaction mixtures were gently stirred overnight at 4 °C. The ^64^Cu chloride was converted to ^64^Cu acetate for radiolabeling by adding an ammonium acetate buffer (pH 6.5) and added to the TETA–SCK solution. The solution was incubated in a thermomixer (1000 rpm) under mild reaction conditions (43 °C for 2.5 h), yielding radiolabeled NPs with a high RCP (>95%) [15].

#### 2.1.3. Discussion

To summarize, many strategies can be used to achieve ^64^Cu-labeled NPs by direct radiolabeling. The possibility of radiolabeling NPs without the use of BFCs is due to their chemical characteristics and, furthermore, this possibility does not apply to all types of NP. For indirect radiolabeling, only TETA was studied for this radioisotope. Since this is a multi-step process, the chemical characteristics of the NPs must not be modified, in order to limit the influence on their targeting specificity as far as possible.

## 3. Radiolabeling of NPs with Alpha-Emitting Isotopes

Alpha particles (helium nuclei) have a high LET, of approximately 80–100 keV/μm, and a penetration range that is typically <100 μm (equivalent to a few cells layer). Due to their high LET, alpha particles mostly interact with DNA, causing irreversible double-strand breaks, and other effects such as G2-phase delay, the generation of reactive oxygen species (ROS), and chromosomal rearrangements [17]. Additionally, β-radiation, for example, can induce damage in neighboring cells [18].

The methods for radiolabeling NPs with α-emitting isotopes are summarized in Table 2.

### 3.1. Radiolabeling with Astatine-211

One of the most promising candidates for targeted α-radionuclide therapy (TAT) is ^211^At. Its long half-life (7.2 h) allows distribution from the production site, labeling procedures, quality control (QC), and clinical application without producing long-lived daughters emitting other α or β particles. Furthermore, this radionuclide is easily produced by cyclotrons with the nuclear reaction of ^209^Bi (α, 2n) ^211^At, followed by dry distillation and isolation from irradiated bismuth [18]. These characteristics make this radionuclide suitable for radiolabeling several types of NP.

#### 3.1.1. Direct Radiolabeling

Astatine is commonly considered a halogen, but the aryl-carbon–halogen bond is very weak, and radiolabeling methods used for other halogens (i.e., iodine) cannot be used without rapid loss, in vivo, of ^211^At from NPs.

To overcome this problem, several authors proposed the use of gold nanoparticles (AuNPs) as carriers for ^211^At. The use of gold is based on the fact that iodine is absorbed by the surfaces of noble metals, establishing strong covalent bonds. Due to the similarities between iodine and astatine, it was expected that ^211^At would form a covalent bond on the surfaces of the AuNPs. For the radiolabeling process, the ^211^At was diluted in Na_2_SO_3_/methanol, the solution was evaporated, and the radionuclide was dissolved in water. For the labeling, 20 mL of ^211^At were added to 200 mL of AuNPs, the pH was maintained at 6, and the final solution was incubated at room temperature under stirring. The radiolabeling yield was >99%, as evaluated by iTLC, with MeOH [19,20,21].

The same method was used for the radiolabeling of gold nanostars (GNSs). In this case, centrifugation was used to separate the free ^211^At from the ^211^At-GNS at the end of the labeling procedure [22].

Kucka et al. investigated the feasibility of radiolabeling silver nanoparticles (AgNPs) using the affinity of astatine for metallic silver. In citrate-phosphate buffers, commercially available AgNPs were labeled at pH 5 and pH 8. Chloramine-T was used to oxidize the ^211^At, followed by reduction with ascorbic acid. The results showed that the pH did not affect the labeling yield, which was greater than 95% in both cases. The same method was then applied to experimentally synthesized AgNPs using KMnO_4_ as an oxidant and NaBH_4_ or Na_2_SO_3_ as a reducing agent. The radiolabeling yield was 77.1% without a reducing/oxidant agent, while rates of 53.7%, 99.7%, and 94.2% were achieved in the presence of KMnO_4_, NaBH_4_, and Na_2_SO_3_, respectively. The use of oxidating or reducing agents can overcome limitations such as the low reproducibility of the radiolabeling yields. The researchers found that astatine can oxidize due to water radiolysis and does not allow it to react on the surface of silver in an oxidation state. The radiolabeling yield and reproducibility can be increased by protecting astatine from oxidation. It was also shown that low water contents in methanol can quench radiolysis and cause it to behave like an oxygen-radical scavenger [23].

#### 3.1.2. Discussion

So far, ^211^At has been radiolabeled with NPs by the direct method only. It is yet to be explored with indirect and encapsulation methods.

The direct radiolabeling of NPs with ^211^At is based on the affinity between the radionuclide and the NPs. The ^211^ At showed a high affinity for gold surfaces, while with silver surfaces, it is essential to prevent astatine oxidation by using a reducing agent in the radiolabeling process.

### 3.2. Radiolabeling with Actinium-225

Actinium-225 *(*^225^Ac) acts as an α-particle generator because of its half-life of 10 days and multiple decaying properties (^221^Fr, ^217^At, ^213^Bi) [41]. The decaying of a radionuclide can occur via a cascade of six radionuclide daughters with short half-lives. This predominantly decaying path generates four α-particles with high energy, while the cascade releases two β disintegrations, such as ^221^Fr and ^213^Bi. The ^213^Bi is an α and β emitter with a half-life of 46 min. It decays to ^213^Po by emitting α particles with half-lives of 4.2 µs, breaking the DNA double strand and causing cytotoxic effects [42].

#### 3.2.1. Direct Radiolabeling

Iron-oxide-based nanoparticles (SPIONs) have attracted broad interest for their intrinsic magnetic properties, like their high surface-area-to-volume ratio and their ability to change their core composition and surface chemically. This chemical modification can be performed through the direct labeling of radioisotopes. Cędrowska et al. synthetized NPs by co-precipitating FeCl_3_ and ^225^Ac, followed by the addition of ammonia to obtain a solution with pH 10. The formed solution was stirred for 15 min, reaching a labeling efficiency of >98% [24].

#### 3.2.2. Indirect Radiolabeling

Single-walled carbon nanotubes (SWNTs) can be radiolabeled indirectly using dodecane tetra-acetic acid (DOTA), a bifunctional chelator.

Ruggiero et al. conjugated DOTA with the primary amines of the SWNT sidewalls. After incubation, ^225^Ac was added to the solution, producing radiolabeled NPs with a 96% RCP [29].

A DOTA-derivative chelator, TADOTAGA, was studied as a BFC for the radiolabeling of AuNPs. The labeling process was similar to that for DOTA. First, TADOTAGA was added to the gold salt solution while stirring, and then a NaBH_4_ aqueous solution was added after a few minutes. After stirring for 1 h, followed by dialysis, the ^225^AcCl_3_ was added and kept in the incubator for 30 min at 70 °C. The sample purification was performed through centrifugation for 15 min at 8110 rpm. The percentage of ^225^Ac was incorporated, and the RCP was determined by iTLC. The results showed RCY values of 86.0 ± 1.8% before centrifugation and 68.5 ± 2.3% after centrifugation, with the RCP greater than 93% [28].

#### 3.2.3. Radiolabeling by Encapsulation

Sofou et al. used liposomes as therapeutic agents for micro metastases and selected the method of encapsulating the radionuclide within the NP. For the synthesis of radioactive liposomes, ^225^Ac was mixed with sucrose-loaded liposomes in PBS and incubated at room temperature for 1h. The binding of the radionuclide to the liposomes was measured using an ultracentrifugation assay, 142,000× *g* for 2 h at 25 °C, allowing the separation of the labeled liposomes from the free ^225^Ac [25].

Inorganic nanoparticles were used as vehicles for targeted radiotherapy. McLaughlin et al. performed a study in which they radiolabeled gold-coated lanthanide phosphate NPs ({La0.5 Gd0.5}(^225^Ac)PO_4_ NPs) and LaPO_4_ (monazite) NPs without BFCs. Firstly, ^225^AcCl_3_ was added to the lanthanide mixture, followed by four additional shells of pure GdPO_4_. The resulting mixture was sealed and heated for 3 h at 90 °C and, finally, purified overnight via dialysis. To test the retention of ^225^Ac decay in vitro, an aliquot was taken from the dialysis tube at different time points and analyzed. The results showed that these NPs were very stable over three weeks, and the NPs retained more than 99.9% of the ^225^Ac [26].

Woodward et al. synthesized LaPO_4_ NPs. In this process, LaCl_3_ in H_2_O was diluted with HCl containing ^225^AcCl_3_. The solution was then evaporated, resuspended in tri-n-butyl phosphate and phenyl-ether, and heated to 100 °C under vacuum with stirring. Next, an aliquot of H_3_PO_4_ solution was added, and the total mixture was heated to 200 °C in an argon atmosphere while stirring for 2 h. Once the solution reached room temperature, the product was separated from the colloids by adding a combination of hexane and hexadecane and centrifuging for 20 min at 14.000× *g*. A sample of NPs was dialyzed for one month to test the in vitro release of the radionuclides. The presence of free ^221^Fr and ^213^Bi was measured in the sample for 2 min at 1-min intervals, while the presence of ^225^Ac was determined by recounting the same sample after 1 h. The results showed the partial sequestration of the ^221^Fr and ^213^Bi daughters, while the ^225^Ac was completely retained in the NPs [27].

Mulvey et al. investigated the radiolabeling of SWCNT with ^225^Ac by three techniques. In the first, ^225^Ac^3+^ ions were added to the NPs; in the second, a mixture of Gd^3+^ ions and ^225^Ac^3+^ ions were added together; and in the third, Gd^3+^ ions, followed by ^225^Ac^3+^ ions, were added sequentially. The samples were sonicated for 2 h to ensure the encapsulation of the ^225^Ac, and then allowed to equilibrate overnight. All the samples were washed and filtered the next day with a paper membrane. The extent of the ^225^Ac^3+^ labeling was determined by serial washing via centrifuge filtration. For the samples with only ^225^Ac^3+^, over 95% of the activity remained with the NPs, while for the other two techniques, only 50% of the activity remained associated, probably due to the excess of Gd^3+^ ions saturating all the binding sites for the ^225^Ac. Challenge experiments in human serum were conducted to simulate in vivo conditions, and the results showed that only 40% of the ^225^Ac^3+^ remained for the US tubes labeled with ^225^Ac^3+^ only [30].

Hernández-Jiménez et al. encapsulated ^225^Ac-radionuclides into reconstituted high-density lipoprotein (rHDL) nanoparticles. The findings demonstrated that the fabricated 13-nanometer nanosystem had more a rate of radiochemical purity of more than 99% and was stable in human serum [31].

Toro-González et al. encapsulated ^225^Ac-radionuclide into gadolinium vanadate (GdVO_4_ NPs). The results showed that the radiochemical yield for the Gd(^225^Ac)VO_4_ was 66.8 ± 7.2% [32].

#### 3.2.4. Discussion

According to published methodologies, the most significant aspect of radiolabeled ^225^Ac-NPs is the stability of the labeled compound over time. The stability of labeling is related not only to ^225^Ac, but also to its decay products [43]. For this reason, materials with high density are required to ensure the sufficient retention of all the decay products within the NPs to limit the cytotoxicity to healthy tissues.

### 3.3. Radiolabeling with Radium-223

As a radium dichloride ([^223^Ra]RaCl_2_), ^223^Ra was the first α-emitting isotope with FDA marketing authorization for the treatment of bone metastases, and it was examined as a potential for targeted radionuclide therapy employing NPs as carriers. This isotope does not have the translocation problem, which is an advantage; while ^225^Ac has a decay chain leading to the generation of daughter nuclides, 75% of the total alphas in the ^223^Ra are delivered within a few seconds (T_1/2_ = 4 s) [44,45,46].

#### 3.3.1. Direct Radiolabeling

The fundamental restriction of ^223^Ra in clinical applications is the difficulty in binding the radioisotope to biomolecules in a stable way. This can be overcome by direct radiolabeling or by encapsulating ^223^Ra into NPs.

The encapsulation of this radioisotope in GdVO_4_ NPs was performed by Toro-Gonzalez et al., with two different methods. The first consisted in mixing a solution containing ^223^Ra evaporated to dryness using an infrared heatlamp and a hot plate with a GdCl_3_ solution and stirred for 10 min, followed by the dropwise addition of a Na_3_VO_4_ solution. In the second method, Na_3_VO_4_ was mixed with the radionuclides, followed by the addition of the GdCl_3_ solution dropwise.

Radiolabeled GdVO_4_ NPs were dialyzed against water to assess the encapsulation and to evaluate the radiochemical yield of the radioisotope and the leakage of the decay daughters. The radiochemical yield was obtained by measuring the initial activity and the activity lost during the synthesis and dialysis. For the first method, the results showed a loss of approximately 25% of the initial activity in the dialysate. With the second method, the results demonstrated a continuous increase in ^223^Ra in the dialysate activity, up to a maximum of 43.6 ± 2.4% for the NPs radiolabeled with the second method. The final radiochemical yields were 34.1 ± 2.9% and 72.3 ± 3.0%, respectively [32].

Liposomes are polymeric NPs can be directly radiolabeled with radionuclides by an ionophore-mediated process. The radionuclide was first added to a vial containing a Ca^−^ionophore film, and the pH was adjusted to 7.4. Liposomes were added to the solution and incubated at 65 °C for 30 min. The addition of enediamine-N and Nl-tetraacetic acid (EDTA) in phosphate-buffered saline (PBS) quenched the loading, and then the mixture was kept in the rest phase for 10 min. Purification was accomplished using size-exclusion chromatography with a PD-10 column. The researchers found that temperature was a significant parameter for the high uptake of the ^223^Ra into the liposomes. The rate of incorporation was <2% at 45 min, and it increased to 78 ± 6% in the final condition [33].

Nanomicelles are colloidal structures with a hydrophilic outer layer and a hydrophobic inner layer formed by amphiphilic monomers. Nanomicells are thermodynamically more stable than conventional micelles, and they are known for their excellent drug-delivery system.

Souza et al. synthesized nanomicelles of [^223^Ra]RaCl_2_ co-loaded with [^198^Au]AuNPs. Non-radioactive AuNPs were synthesized and irradiated for 12 h in an Argonauta reactor to produce [^198^Au]-AuNPs. Subsequently, the [^223^Ra]RaCl_2_ nanomicelles were synthesized, and a Pluronic F127 copolymer was dispersed while stirring. Next, [^198^Au]AuNPs were added to the solution and ultrasonicated for more than 2 min in an ice bath at 10 °C. The in vitro citotoxicity studies revealed a substantial increase in tumor-cell death when the treatments with both [^223^Ra]RaCl_2_ and [^198^Au]Au-NPs were combined in the same formulation, compared to the [^223^Ra]RaCl_2_ or [^198^Au]Au-NPs alone [35].

Iron-oxide NPs can be radiolabeled without with a chelator utilizing a heat-induced metal-ion-binding method, which avoids the use of a multi-step radiolabeled process by using a diagnostic radioisotope in the indirect approach [4].

Gholami et al. developed radiolabeled NPs with superparamagnetic iron oxide (SPIO) cores, commonly known as Feraheme^®^ (FH) NPs, in which radioisotopes bind directly to the surface of the SPIO core. The therapeutic radioisotopes used were ^223^Ra, ^177^Lu, ^90^Y, and ^64^Cu. The study showed that the FH NPs could be radiolabeled with high RCY and RCP using both diagnostic radioisotopes with long-range beta emitters and therapeutic radioisotopes with short-range alpha emitters [36].

Gawęda et al. radiolabeled barium-ferrite nanoparticles (BaFe NPs) with ^223^Ra. Iron chloride and barium chloride were mixed in an 8:1 molar ratio (Fe^3+^:Ba^2+^), and then ^223^Ra chloride salt was added, followed by sodium hydroxide. The percentage of ^223^Ra incorporated was calculated from the ratio of the radioactivity retained inside the nanoparticles to the radioactivity of the ^223^Ra initially added, producing 61.3 ± 1.8% [37].

Suchánková et al. compared two different approaches to radiolabeling with hydroxyapatite NPs. The radionuclide was directly coupled with the surfaces of previously synthesized NPs in the first approach. After dispersing the NPs in a physiological saline solution, ^223^Ra was added. The samples were mixed for 1 h at room temperature and then washed thrice with saline. The second approach was used to incorporate radionuclides into the structures of nanoparticles by adding Ca(NO_3_)_2_ to demineralized water. Both methods showed high RCY (≥94%), demonstrating that they are both feasible strategies for labeling, with the caveat that the daughter radionuclides were not determined [38].

Graphene quantum dots (GQDs) were chosen from among carbon-based nanomaterials for direct radiolabeling with ^223^Ra. The radiolabeling process was simple; after the addition of EDC in a Britton–Robinson buffer in a GQD-FA solution, an aliquot of [^223^Ra]RaCl_2_ was added, and the mixture was heated under stirring at 90 °C for 1 h. To demonstrate the presence of radiolabeled NPs, X-ray spectroscopy (EDS), was performed after two and fourteen days of radiolabeling, associating the presence of ^223^Ra with the formation of ^211^Pb and ^207^Pb as products of ^223^Ra decay [39].

Piotrowska et al. radiolabeled nanozeolites with ^223^Ra by exchanging the Na^+^ with the ^223^Ra^2+^ cations in the zeolite structure. A ^223^Ra(NO_3_)_2_ solution was added to the nanozeolite sample, which was then sonicated in an ultrasound bath for 15 min and gently shaken for 2 h. Next, the suspension was centrifuged at 13,000 rpm for 10 min and separated from the supernatant. The findings suggested a successful radiolabeling, with a high affinity and stability of ^223^Ra [34]. Mokhodoeva et al. radiolabeled superparamagnetic iron-oxide nanoparticles (SPIONs) with ^223^Ra and reported an analysis of the ^223^Ra uptake by the Fe_3_O_4_ SPIONs. The results showed that the uptake speed of the radium was moderate, and 30–60 min were required for the radiolabeling with SPIONs, with yields of over 85%, ranging up to 99% [40].

Subsequently, the same research group studied the release of ^223^Ra, ^211^Bi, and ^211^Pb recoils from radiolabeled NPs [45]. The results showed the low release of all the isotopes from the NPs, depending on the method used to separate them (dialysis or centrifugation). With these experiments, the authors highlighted that false-positive results of radiolabeled NP stability can occur and, in particular, the recoiling progeny may induce artifacts. In conclusion, the authors suggested that in vitro stability tests should always be designed under real or very-similar-to-real conditions [45].

#### 3.3.2. Discussion

The radiolabeling of ^223^Ra with NPs using BFCs is a complicated process due to the properties of the radioisotope; this incorporation can be achieved by the direct radiolabeling method. Studies highlight strategies that allow direct radiolabeling; however, certain restrictions are associated. For example, Radon-219 (^219^Rn) is a daughter isotope of ^223^Ra, whose in vivo redistribution is unknown. Further investigation into the biodistribution of ^223^Ra and its daughter isotopes is required for future clinical application.

## 4. Radiolabeling of NPs with Beta-Emitting Isotopes

Beta (β) emitters are the most commonly used radionuclides in therapy. These β-emitting radioisotopes have the largest particle range (≤12 mm) and lowest linear energy transfer (LET) (0.1–1.0 keV μm^−1^). The β particles emit electrons that interact with atoms by passing through tumor tissues and losing their energy, leading to the release of ionized and excited atoms, as well as free radicals, which cause single-strand breaks in DNA and cell damage. However, this damage can also affect healthy tissue around tumor sites [46,47,48]. The methods for radiolabeling NPs with alpha-emitting isotopes are summarized in Table 3.

### 4.1. Radiolabeling with Gold-198

The radionuclide Au-198 (^198^AuCl_4_) has a half-life of 2.7 d with cyclotron production and β and γ decay. The main advantage of the application of this isotope in nanomedicine is that the radiochemical compound is often the same element of the NP structure (AuNPs).

#### 4.1.1. Direct Radiolabeling

Extensive studies of AuNPs have been carried out for a wide range of applications. Their characteristics, such as their easy synthesis process, the possibility of their functionalization, their inertness, and their peculiar optical properties, make them interesting candidates for both diagnostic and therapeutic purposes [114]. Indeed, most studies used AuNPs to synthesize ^198^AuNPs for radiation therapy (RT) with the goal of delivering a lethal dosage of radiation to tumor lesions while avoiding exposure to surrounding healthy tissues. The synthesis of ^198^Au-labeled NPs differs significantly from that with other isotopes. Most investigations are based on the neutron irradiation of the gold foil, which produces radiolabeled NPs. The gold foil is irradiated by a neutron flux from a ^198^Au reactor, and then dissolved in aqua regia, followed by the addition of HCl, after which it is allowed to evaporate the solvent.

This evaporation process is repeated three times, and then the specimen is left at room temperature for the crystallization of H^198^AuCl_4_·3H_2_O. Once the radioactive precursor is synthesized, the next step is conjugation with additional molecules to improve the specificity of the NPs.

Several molecules are used for this purpose: arabinoxylan (AX), a hemicellulosic material with a colon-specific uptake; gum arabic glycoprotein (GA), a plant extract approved by the FDA to target tumor cells; epigallocatechin-gallate (EGCg), which was shown to be beneficial in treating brain, prostate, cervical, and bladder cancers; mangiferin (MGF), a prostate-tumor-specific natural substance; cyclic arginine−glycine−aspartate peptide (RGD), known as a peptide that can specifically bind to integrin-αvβ3 expression in tumor angiogenesis; and, finally, human serum albumin (HSA) or apolipoprotein E (apoE).

Conjugation with these molecules often requires a reducing agent, such as N_2_H_4_ or NaBH_4_, to reduce Au^3+^ ions to Au^0^, stabilizing the isotope.

The characterization of NPs can be performed in different ways: by surface plasmon resonance (SPR), in which absorption at 526 nm is used to determine the shape and the particle size; by transmission-electron microscopy (TEM); by ultraviolet-visible spectrophotometry, in which an absorption band around 540 nm is characteristic of AuNPs; by high-resolution γ-ray spectrometry using a HPGe detector; or by nuclear magnetic resonance (NMR).

The particle size is frequently measured using dynamic light scattering (DLS), and a DLS analysis of AuNPs revealed the formation of very small nanoparticles (12–18 nm), while Zetasizer Nano ZS confirmed the polydispersity index (PDI) of 0.106 [35,48,49,50,51,52,53,54].

The surface charges of nanoparticles are an important factor in influencing their biological properties. The biodistribution of AuNPs is affected due to surface charge, such as in the kidneys, positively charged particles are used to accumulate, while a high accumulation of negative and non-charged particles was shown in the liver [15]. Hirn et al. radiolabeled ^198^Au with 1.4-, 5-, 18-, 80-, and 200-nanometer negatively charged AuNPs and 2.8-nanometer positive charged AuNPs. The biodistribution of AuNP was measured by gamma spectrometry after 24 h of intravenous administration into rats. The results showed that the liver accumulation increased from 50% of 1.4-nanometer AuNPs to more than 99% of 200-nanometer NPs, while most other organs had small size-dependent accumulations of 18–200-nanometer AuNPs. However, the accumulation of AuNPs between 1.4 nm and 5 nm increased significantly with a decrease in the particle size, showing a linear relationship with the volumetric specific surface area [115,116].

The development of both nanobiological interfaces and their potential treatment requires an understanding of the in vivo effects of the surface chemistry of AuNPs. Wang et al. investigated the in vivo biodistribution, excretion, and toxicity of three different surface-charged glutathione-protected gold nanoclusters (AuNCs), but with equal hydrodynamic sizes. In addition, the authors analyzed the potential application of these nanoclusters in radiation therapy and tumor uptake. The findings revealed that the surface charge significantly affected the pharmacokinetics, specifically in terms of their renal excretion and accumulation in the liver, testes, kidneys, and spleen. The peripheral blood system was temporarily affected by positively charged clusters, while negatively charged Au NCs had lesser excretion and higher tumor uptake, implying the potential of NC-based therapeutics [117].

#### 4.1.2. Discussion

As discussed above, a research reactor is needed to synthesize ^198^Au-labeled NPs. This approach allows radiolabeling with high radiochemical yields, but it can only be used during the synthesis of NPs.

### 4.2. Radiolabeling with Holmium-166

Holmium-166 (^166^Ho) is a radioisotope with a half-life of 26.9 h; it is used in gamma scintigraphy by combining the emission of therapeutic β^−^ particles with γ rays [118].

These features make ^166^Ho an ideal isotope for theragnostic purposes. Furthermore, ^166^Ho is a paramagnetic and electron-dense element that can be used for X-ray computed tomography (CT) and magnetic resonance imaging [119,120,121].

#### 4.2.1. Direct Radiolabeling

Holmium-acetylacetonate microspheres (HoAcAcMS) were studied by Bult et al. Once synthetized, the samples were irradiated with a thermal neutron flux of 5 × 1012 cm^−2^ s^−1^ for 3 or 6 h, and the radioactive ^166^Ho was left to decay closed in vials for one month. The degradation of the microspheres (the release of Ho^3+^ ions from HoAcAcMS) was evaluated by suspending 100 μL of ^166^HoAcAcMS in 2% Pluronic^®^ F68 aqueous solution and incubation in test tubes containing an isotonic phosphate buffer. The test tubes were placed in a continuously shaken water bath at 37 °C and centrifuged at different times (1 day, 4 days, and 1, 2, 4, 8, 12, and 24 weeks). The supernatant was analyzed after six months of incubation in the buffer and showed a release below 0.5%, demonstrating the stability of ^166^HoAcAcMS. This was also confirmed in in vivo, by studies carried out in VX2-tumor-bearing rabbits, where no release of holmium was observed, and the microspheres remained intact in the tumor tissue for one month [111].

Electrospinning is a simple and low-cost technology for producing nanoparticle-embedded fibers with homogeneously dispersed NPs. Munaweera et al. incorporated non-radioactive holmium-165 (^165^Ho) iron-garnet nanoparticles (HoIGNPs) into a bandage by using the electrospinning method for neutron irradiation. The polymer fibers were stable following neutron activation, with no leakage of radioactive material [112].

The radiolabeling process can also be performed after synthesizing nanoparticles without using a chelator. Vimalnath et al. radiolabeled agglomerated IONPs by mixing them with a ^166^HoCl_3_ solution (obtained by radiative neutron capture) and stirring the mixture at room temperature for 60 min. The ^166^HoIONPs were separated from the supernatant, after which the unlabeled residue was purified by washing with a sterile solution. The findings demonstrated that a minimum amount of IONPs (5 mg of particles in 1 mL of reaction volume) achieved a radiolabeling yield of more than 95%. The yield was excessively low at pH levels below 5, whereas no significant changes were detected at pH levels between 5 and 10. As a result, the optimal pH was determined to be 7–8, which is advantageous, since it is close to the physiological pH [113].

#### 4.2.2. Discussion

We reported the neutron irradiations and the electrospinning approach to radiolabel NPs with ^166^Ho. The neutron-irradiation method is effective but time-consuming, whereas electrospinning is simple and cost-efficient. However, both techniques led to a high radiochemical yield with a non-significant release of free ^166^Ho over time.

### 4.3. Radiolabeling with Yttrium-90

Yttrium-90 (^90^Y) is a β-emitter with a half-life of 64.1 h and a high average level of energy (2.27 MeV), making it ideal for radionuclide therapy for large tumors, since it can damage tumor cells up to a depth of 11 mm in soft tissue [122]. Nevertheless, significant damage can also occur to neighbor normal tissues.

#### 4.3.1. Direct Radiolabeling

Magnetic nanoparticles (MNPs) are found to be suitable for direct radiolabeling with ^90^Y. Radović et al. synthesized Fe_3_O_4_-MNPs and Fe_3_O_4_ functionalized polyethylene glycol 600 diacid (PEG600)-MNPs to compare their efficacy as radioactive vectors. The radiolabeling was achieved by mixing the NPs solution with ^90^YCl_3_ and stirring the reaction mixture for 1 h at room temperature. The results showed high radiolabeling yields for both MNPs: 97% for the naked Fe_3_O_4_ and 99% for the Fe_3_O_4_–PEG600 [55].

Ognjanovíc et al. used ^90^Y and ^177^Lu with IONP-coated citric acid (CA), poly(acrylic acid) (PAA), and poly(ethylene glycol) (PEG). The radiolabeling process was based on incubating the radionuclide with the IONPs for 30 min or 60 min. The iTLC confirmed that the radiolabeling yields were greater than 99% for every sample except for the ^177^LuPEG-IONP, regardless to the incubation time [56].

In another work, Radović et al. designed ^90^Y-labeled imidodiphosphate- and inositol-hexaphosphate-coated magnetic nanoparticles. Their iTLC analysis confirmed that the radiolabeling yield from the coated-MNPs was greater than 98% [57].

Soni et al. radiolabeled YPO_4_:Er^3+^−Yb^3+^ NPs, rare earth (RE) NPs, with ^90^YCl_3_ or ^177^LuCl_3_, without the use of any BFC. The binding equilibrium for both radioisotopes was achieved within 30 min of incubation at pH 6.5; after the purification, both formulations achieved a radiochemical purity of more than 95% [58].

#### 4.3.2. Indirect Radiolabeling

Diethylenetriaminepentaacetic acid (DTPA) and DOTA are primarily used, as BFCs, for radiolabeling NPs with ^90^Y.

Li et al. used DTPA for the radiolabeling of liposomes. The liposomes were conjugated with DTPA in a chloroform solution. The chloroform was then removed by evaporation at 60 °C, water was added to obtain an aqueous solution, and ^90^YCl_3_ was added at room temperature for 30 min. The RCY was evaluated using a nanosep filter in a microfuge at 4000 rpm, and the results showed that more than 95% of the ^90^Y bound to the liposomes [59].

Buckway et al. used a N-(2-hydroxypropyl)-methacrylamide (HPMA) copolymer to synthesize polymeric NPs and radiolabeled them with DOTA. The HPMA-DOTA copolymerization was achieved using reversible addition–fragmentation-chain transfer (RAFT), after which ^90^YCl_3_ was added, and the solution was incubated for 1 h at 50 °C under nitrogen. EDTA was added to the solution after it reached room temperature to eliminate any free or loosely bound ^90^Y^3+^ ions. The results showed that DOTA may be a stable chelator for this radioisotope [60].

#### 4.3.3. Discussion

As discussed above, the negative surface charging of nanoparticles is required for direct radiolabeling with positively charged ^90^Y^3+^. If the native NPs do not have a negative charge, they can be coated with multiple molecules without affecting their chemical–physical properties. Furthermore, different BFCs can effectively be used for indirect radiolabeling.

### 4.4. Radiolabeling with Lutetium-177

Lutetium-177 (^177^Lu) has a half-life of 6.7 days; it is considered ideal for the theragnostic emission of γ photons and β^−^ particles, which penetrate only 2 mm into tissues and, thus, spare normal neighboring tissues or cells from irradiation. In addition, ^177^Lu has been used in radio-synovectomy and hepatocellular carcinoma therapy, and it is currently used for several new treatments for cancers, including neuro-endocrine tumors [123] and prostate cancer [124].

#### 4.4.1. Direct Radiolabeling

By modifying the surfaces of MNPs, to design potential theragnostic agents, these can be radiolabeled as chelator-free with high radiochemical purity and stability. Salvanou et al. radiolabeled alginic coated (MA) IONPs with ^177^Lu (^177^Lu–MA) and also stabilized them with a layer of polyethylene glycol (PEG) (^177^Lu-MA-PEG). The iTLC-SG analysis confirmed values of 95.2 ± 1.3% RCY for the ^177^Lu-MA and 93.6 ± 1.0% for the ^177^Lu-MA-PEG [61].

Poly-L-lysine, proline, and tryptophan can also be used for MNP coating. Mirković et al. fabricated poly-L-lysine coated MNPs (PLL-MNPs) and modified IONPs with proline (Pro-MF) and tryptophan (Trp-MF). These NPs were radiolabeled with two therapeutic radioisotopes, ^177^Lu and ^131^I. The radiolabeling yield was quantified by iTLC, which showed that the highest radiolabeling yield was for the PLL-MNPs (99% for both formulations), while the Pro-MF and Trp-MF yielded 59.5% and 45.6%, respectively, at room temperature, and 69.2%, and 70.3% at 80 °C [62].

Phosphates like imidodiphosphate (IDP) and inositol hexaphosphate (IHP), and phosphonates like methylene diphosphonate (MDP) and hydroxyethylidine diphosphonate (HEDP), were also used as coatings for MNPs.

Radović et al. radiolabeled ^177^Lu with both phosphates and phosphonate-coated MNPs. Their iTLC analysis confirmed a radiolabeling yield of 99% from the ^177^Lu-labeled phosphate-coated MNPs, and yields of 78.7 ± 0.7% and 75.0 ± 1.1% for the ^177^Lu-MDP-MNPs and ^177^Lu-HEDP-MNPs, respectively. The radiolabeling yield increased to 95% after the purification of the phosphonate-coated MNPs [63].

Joshi et al. recently published an interesting report on the radiolabeling of NaGdF_4_/Ho-Yb@m-SiO_2_ nanoparticles (UCNPs) with ^177^Lu, obtaining 98.4 ± 0.4% RCY, and studied their biodistribution in rats, showing no uptake of free ^177^Lu in bones up to 168 h [64].

Chakraborty et al. radiolabeled hydroxyapatite NPs with ^177^Lu. Their results showed a high radiolabeling yield (99.1 ± 0.2%), and no radioactivity of the free radioisotope was detected [66].

Finally, Azorín-Vega et al. radiolabeled Tyr3-octreotate-conjugated AuNPs with ^177^Lu. Their iTLC analysis confirmed radiochemical purity values of more than 98% [90].

#### 4.4.2. Indirect Radiolabeling

Among the BFCs, ^177^Lu can form a stable coordination complex with DOTA or with its derivates. The radiolabeling process can be performed either by labeling NPs with ^177^Lu-BFC or by labeling BFC-conjugated NPs.

For the first strategy, the radiolabeling of DOTA with ^177^Lu required an incubation time at a high temperature (80–90 °C) and a purification step to remove the unbound ^177^Lu. Next, conjugation with AuNPs was performed in mild conditions under stirring for a few minutes. The results showed that DOTA can efficiently bind to AuNPs. Indeed, the radiochemical purity in all the published papers was >90% [84,85,86,87,88,89,90].

The second strategy was chosen by several other authors, who chose a very similar methodology.

The DOTA was not only used with AuNPs but also with several other types of NP, such as PLGA-NPs. To conjugate the chelator, lyophilized PLGA-NPs were dissolved in water, DMF, DIPEA solution, and carboxylate-activating agent (HATU). The DOTA was added subsequently, and the mixture was incubated under slow stirring for 2 h at room temperature. Next, a solution of ^177^LuCl_3_ was added in an acetate buffer (pH 7) to the DOTA-PLGA-NP solution and was then heated at 37 °C for 3 h, producing radiolabeled NPs with high radiochemical purity [70,71].

Mendoza-Nava et al. attached ^177^Lu to a DOTA-derivate, S-2-[4-Isothiocyanatobenzyl]-1,4,7,10-tetraazacyclododecane tetraacetic acid (p-SCN-benzyl-DOTA)-conjugated-G4-PAMAM-(NH_2_)64 dendrimer. The size-exclusion radio-HPLC confirmed a level of radiochemical purity of more than 95% [73]. Two further examples were published by Mendoza-Nava et al. [69] and by Kovacs et al. [72]. Here, DOTA was bound to polyamidoamine (PAMAM) dendrimers, followed by an incubation of 1–1.5 h, and, subsequently, by the addition of ^177^LuCl_3_, and incubated at 37–50 °C for 20–60 min.

In addition, DOTA can be conjugated with NPs during the synthesis of particles, as demonstrated by Wang et al. when they radiolabeled DOTA-tri arginine-lipid with ^177^Lu [75].

Finally, Goos et al. functionalized a nanostar polymer (p(AEA-co-OEGA-co- [Gd^3+^]VDMD) with trans-cyclooctene (TCO) to enable the conjugation of nanostars with tetrazine-functionalized DOTA (Tz-DOTA). This two-step labeling approach was chosen to prevent the metal exchange of Gd^3+^ and ^177^Lu^3+^. First, the Tz-PEG7-DOTA was radiolabeled with ^177^Lu^3+^, and then it was reacted with star polymer to obtain a p([^177^Lu]Lu-DPAEA-co-OEGA-co-[Gd^3+^]VDMD) star polymer, with a high radiochemical yield (87%) and purity (>99%) [76].

Reconstituted high-density lipoproteins (rHDLs) are promising platforms for developing theragnostic systems due to their ability to transport and release drugs and imaging agents through the scavenger receptor (type B1 (SR-B1)) to tumor-targeting sites. Aranda-Lara et al. studied ^177^Lu-radiolabeled rHDLs, using DOTA as a chelator. The radiochemical yield was 85% before the purification (performed by centrifugation) and 99% after the purification. Moreover, these NPs showed high stability, with a radiochemical purity >95% after 24 h of incubation at 37 °C in saline and human serum [77].

Vats et al. developed ^177^Lu-labeled cyclic Asn–Gly–Arg peptide-tagged carbon nanospheres as potential radio-nanoprobes. The radiochemical yield was confirmed by an iTLC analysis at 80% [78].

Zhang et al. fabricated alpha-melanocyte-stimulating hormone (αMSH)-surface-engineered PEG-coated silica NPs and radiolabeled them with ^177^Lu by using DOTA as a chelator to target the melanocortin-1 receptor (MC1-R) in melanoma-tumor cells. The results showed a radiochemical yield of more than 95% [79].

Shultz et al. investigated the theragnostic effect of radiolabeled ^177^Lu and ^177^Lu-DOTA with functionalized metallofullerene (Gd_3_NC80) NPs (^177^Lu-Gd_3_NC80 and ^177^Lu-DOTA-f-Gd_3_NC80) in brachytherapy. The results showed radiopurity of 80%, while the in vivo stability study revealed some ^177^Lu-DOTA degradation or cleavage from the f-Gd_3_NC80 surface. Despite this level of in vivo degradation, this investigation demonstrated effective brachytherapy by comparing ^177^Lu-labeled f-Gd_3_NC80 to f-Gd^3^NC80 alone [80].

D’Huyvetter et al. designed ^177^Lu-nanobodies by conjugating DOTA and DTPA chelators to target HER2-positive breast cancer. Both formulations showed radiochemical purity of more than 98% in the iTLC analysis [81].

In addition, DTPA was used for liposomes. The conjugation between the chelator and the NPs was performed during the synthesis of the liposomes, which involved mixing the lipids in chloroform-methanol and dimyristoylphosphoethanolamine-DTPA (DMPE-DTPA) in two different molar ratios to achieve two formulations, with “high” DTPA loading and with “low” DTPA loading. The radiolabeling was performed by simply adding a solution of ^177^LuCl_3_ to the liposomal solutions and incubating the solutions at room temperature for 30 min. The RCY was higher than 80%, but a high amount of NPs was necessary. This was due to the higher amount of carrier-lutetium in this radionuclide, which resulted in a higher mass of lutetium per radioactivity, which led to achieving a lower SA [82].

Ge et al. reported that Fe_3_O_4_ NPs can be labeled with ^177^Lu using DP-PEG as a chelating agent. They also labeled these NPs with ^99m^Tc for SPECT and with ^68^Ga for PET imaging. The RCY was approximately 50%, but the labeling was stable and did not modify the physical properties of the NPS [67].

Finally, ^177^Lu-labeled polyaminoamide (PAMAM) dendrimers using a DOTA-like bifunctional chelator and methylenepyridine-N-oxide pendant arm (DO3A-py(NO-C)) showed a high labeling yield, stability, and a rate of chemical purity of over 98% [74].

#### 4.4.3. Radiolabeling by Encapsulation

The encapsulation of radioisotopes in nanoparticles may prevent biological degradation, and, as a result, lower concentrations may be sufficient to achieve the desired effects. Chakravarty et al. developed a human serum albumin (HSA)-mediated biomineralization process for the synthesis of intrinsically radiolabeled ^177^Lu_2_O_3_ nanoparticles entrapped in a protein scaffold (^177^Lu_2_O_3_-HSA nanocomposite). The radiochemical purity of the ^177^Lu_2_O_3_−HSA nanocomposite was determined by radio-thin-layer chromatography (radio-TLC), and a purity rate of more than 98% was achieved [65].

Arora et al. fabricated ^177^Lu-DOTA-TATE-loaded PLGA nanoparticles (NPs). The results showed that the encapsulation efficiency (EE) was between 77.3 ± 4.9% and 58.4 ± 5.3%, depending on the experimental condition. The authors showed that PLGA-NPs also have a slower release rate for the isotope [83].

#### 4.4.4. Discussion

The direct method of radiolabeling NPs with ^177^Lu requires high temperatures and may disrupt the NPs; on the other hand, the indirect method may take a long time under acidic pH [124]. To overcome these limitations, the radiolabeling of NPs can be performed with ^177^Lu-DTPA. The encapsulation approach for radiolabeling is preferable because of its excellent yield and reproducibility under mild and physiological conditions.

### 4.5. Radiolabeling with Iodine-131

It is known that ^131^I, which has a radioactive decay half-life of 8.02 days, is the most commonly used β-particle-emitting radioisotope in nuclear medicine for treating thyroid cancer, neuroblastoma, and pheochromocytoma [125].

The emission of γ-rays makes it a potential radioisotope for molecular imaging [3]. The radiolabeling of NPs with iodine radioisotopes is based on halogen-nucleophilic exchange, which can be conducted directly with an oxidizing agent via the direct radioiodination of NPs, or indirectly, by employing prosthetic groups.

#### 4.5.1. Direct Radiolabeling

Chen et al. radiolabeled palladium (Pd) nanosheets with ^131^I by using the direct method without using an oxidizing agent. The results showed a labeling efficiency of more than 98% [110].

Among the oxidizing agents, the 1,3,4,6-tetrachloro-3α,6α-diphenylglycouril (Iodogen) method and the N-chlorobenzenesulfonamide (chloramine-T) method are the frequently most used. The first is a less aggressive process based on the dissolution of Iodogen in an organic solvent and evaporated in the presence of a tube, remaining bound to the wall. The second is a strong oxidant that leads to radioiodination in a few seconds, but a reducing agent is required to stop the reaction. To control the oxidizing activity without affecting the biomolecules present in the solution, polystyrene beads are used to fix the activity; thus, the oxidizing reaction is stopped when the beads are removed from the solution [126].

The Iodogen method allows the radiolabeling of several types of NPs. İnce et al. synthesized the glucuronide derivative of thymoquinone (TQG) enzymatically, conjugated it with a synthesized MNP, and then radio-iodinated the mixture with ^131^I. The findings suggested that the nanoparticles were radio-iodinated with ^131^I with a yield of over 95% [92].

Er et al. determined intracellular uptake and in vivo photodynamic-therapy potential of Zn phthalocyanine-loaded ^131^I-labeled mesoporous silica nanoparticles (MSNP5) against pancreatic cancer cells. The radiolabeling efficiency was 95.5 ± 1.2% with a pH of 9 and a 60-min reaction time. Furthermore, the highest intracellular-uptake yields of the ^131^I-MSNP5 nanoparticles in the MIA PaCa-2, AsPC-1, and PANC-1 cells were determined as 43.9 ± 3.8%, 41.8 ± 0.2%, and 37.9 ± 1.3%, respectively, with 24 of incubation [93].

He et al. incorporated ^131^I into a PAMAM (G5.0) dendrimer to target thyroid carcinoma. The radiolabeling efficiencies of the ^131^I–PAMAM (G5.0) and ^131^I–PAMAM (G5.0) were 93 ± 1% and 85 ± 2%, respectively. The ^131^I–PAMAM (G5.0) exhibited small but significant changes in radiochemical purity as a function of time after labeling. The highest observed purity was 82 ± 2%. The ^131^I–PAMAM (G5.0)–VTP displayed larger changes in radiochemical purity as a function of time after labeling, with a maximum of 80 ± 2% [94].

Avcıbaşı et al. radiolabeled N-methacryloyl-l-phenylalanine (MAPA) containing poly(2-hydroxyethylmethacrylate) (HEMA)-based magnetic poly(HEMA–MAPA) nanobeads [mag-poly(HEMA–MAPA)] with ^131^I [^131^I-mag-poly(HEMA–MAPA)]. The binding yield of the ^131^I-mag-poly(HEMA–MAPA) was about 95–100% [91].

Zhu et al. synthesized a G5.NH_2_ PAMAM dendrimer modified with 3-(4′-hydroxyphenyl) propionic acid-OSu (HPAO) and folic acid (FA) linked with polyethylene glycol (PEG) and labeled with ^131^I. An iTLC was performed to evaluate the radiochemical purity and stability at different time points (0–1–3–27 h). The results showed a radiochemical purity of 97.7%, which remained at >90% on each occasion [95].

Zhang et al. synthesized ^131^I-labeled bovine serum albumin (BSA)-modified copper sulfide (CuS) NPs (^131^I-BSA@CuS), with attributes of both radiotherapy and PTT, as a therapeutic agent against anaplastic thyroid carcinoma (ATC). Their iTLC analysis confirmed a radiochemical purity 66–80% [103].

Chen et al. developed albumin-based gadolinium-oxide NPs (GdNPs) loaded with doxorubicin (DOX) and conjugated with bone-seeking alendronate (ALN) to treat malignant bone destruction. The abundance of tyrosine residues in the albumin molecule resulted in a high radiolabeling efficiency [107].

Using the chloramine-T method, Zhao et al. radiolabeled ^131^I to the surface of polydopamine (PDA)-coated SWCNT modified with PEG to increase blood-circulation half-life (SWCNT@PDA-PEG). The results showed a radiolabeling yield of approximately 90% [97]. Several other authors used the chloramine-T method [99,100,102,103], with RCY values varying from 55% to 90%, suggesting the need for the further purification of radiolabeled NPs from unbound iodine. As an alternative to chloramine-T, Liu et al. used NaClO_4_ as an oxidizing agent to oxidize iodine from I^−^ to I^+^. They labeled hollow copper-sulfide NPs with paclitaxel, thus combining phototermal, chemo- and radio-therapies [109].

#### 4.5.2. Indirect Radiolabeling

The Bolton–Hunter reagent, the N-hydroxysuccinimide ester of iodinated p-hydroxyphenyl propionic acid, is the only BFC utilized for iodine radioisotopes. It is used to iodinate molecules that do not have tyrosine residues, allowing them to be radiolabeled with NPs.

Its precursor, 3-(4-hydroxyphenyl)propionic acid-OSu (HPAO), was also used to radiolabel NPs, particularly for the radiolabeling of polyethyleneimine-entrapped gold nanoparticles (Au PENPs) and PAMAM dendrimers. For the former, polyethyleneimine (PEI) was selected as a template for the further modification with PEG, chlorotoxin, a glioma-specific peptide (CTX), and HPAO, which were then used to entrap gold nanoparticles (Au NPs). Conjugation with HPAO was achieved by adding HPAO to the template after the modification with PEG and CTX, and stirring overnight. After trapping the AuNPs in the modified PEI,PENP-CTX was obtained. The radiolabeling process was performed by, first, mixing the Na^131^I solution with chloramine-T and PENPs-CTX with continuous stirring. The final mixture was incubated for 30 min at 37 °C and was then purified with a PD-10 desalting column. The radiochemical purity (determined by ITLC) was measured at different times to evaluate the radio-stability, showing a radiochemical purity of over 99%, remaining above 90% after 24 h in PBS at room temperature and in FBS at 37 °C [97,99].

Song et al. synthesized ^131^I-labeled dendrimers modified with the LyP-1 peptide as a multifunctional platform for single-photon emission computed tomography (SPECT) imaging and radionuclide therapy for metastatic cancer. The iTLC analysis confirmed a radiochemical purity of more than 99% and radiochemical stability of more than 90% [96].

Finally, Ji et al. radiolabeled albumin nanospheres loaded with doxorubicin using a new chelating agent, namely N-succinimidyl-3-(2-pyridyldithiol)propionate (SPDP). Firstly, iodine was bound to a monoclonal antibody (anti-AFPMc), and then the radiolabeled antibody was linked to the NP via the SPDP cross-linker. It was not possible to use the resulting construct to perform chemo–radiotherapy for liver cancer. The results obtained from Balb/c mice with HepG2 xenografts showed improved tumor reduction compared with mice treated with doxorubicine alone [100].

#### 4.5.3. Radiolabeling by Encapsulation

The chloramine-T method was used for labeling BSA encapsulated into the cores of liposomes. The encapsulation of the radionuclide was possible due to the capacity of the oxidized ^131^I_2_ to pass through the phospholipid bilayer of the liposome and react with the tyrosine residue present on the BSA. The oxidation was carried out by mixing the liposomes with K^131^I and chloramine-T. The solution was vortexed for 10 min, and then purified using a Sephadex G-100 column. The radiolabeling yield, measured by a gamma counter, was over 90% [106].

Sakr et al. synthesized polyethylene-glycol-capped silver nanoparticles doped with I-131 radionuclide (^131^I-doped Ag-PEG NPs). The radiolabeling yields and the encapsulation efficiency of the Ag-PEG NPs were studied at room temperature for up to 8 h. The maximum radiolabeling yield (98 ± 1%) was obtained after 3 h of stirring, and there was no ^131^I-leakage from the Ag-PEG NPs until 6 h post-synthesis [108].

#### 4.5.4. Discussion

The radiolabeling of NPs with ^131^I typically involves the electrophilic aromatic substitution of phenolic or trialkylstannylated substrates using an oxidant agent, such as chloramine-T or Iodogen [127]. With these methods, high radiochemical yields are obtained. The main limitation of iodine-labeled NPs is deiodination, with the undesired accumulation of radionuclides in normal tissues, such as the thyroid and stomach. Therefore, the fabrication of stable and multifunctional nanocarriers for the efficient delivery of radioactive iodine to targets is required [127].

## 5. General Conclusions

There has been a significant increase in interest in integrated research at the interface of nuclear medicine and nanotechnology over the last two decades [126,128]. This developing convergent science can potentially overcome the limitations of current radionuclide therapy. This review highlights the radiolabeling of NPs with Auger’s electron-emitting isotopes, alpha-emitting isotopes, and beta-emitting isotopes by direct, indirect, and encapsulation methods. Nanostructures with linked Auger’s electron emitters have a substantial advantage over traditionally labeled biomolecules, delivering significantly more radioactivity to targeted tissues. This is important for Auger’s electron therapy since it needs considerably higher radionuclide activity to achieve a therapeutic effect. Therapeutic β-emitting radioisotopes are found in various organic, inorganic, and hybrid forms of nanomaterials. Radiolabeling is typically accomplished through the use of a bifunctional chelator or through the formation of covalent bonds. The easy availability of β-emitters and successful results may enable the development of various nanoplatforms for effective theragnostic research. The α-particle-incorporated nanomaterials have received less attention because of their higher production costs and lower availability. The encapsulation of radionuclides within nanostructures was preferred for generating stable therapeutic agents because commonly employed chelating agents or labeling processes typically cannot sequester the α-emitter and its decay products.

Nanoparticles can be radiolabeled with radioisotopes by multiple methods to allow multimodal therapy. The specific therapeutic application of NPs is based on the most appropriate choice of nanomaterials and isotopes, and the most suitable radiolabeling method. Indeed, all these factors influence the results and the possible application of the method to humans.

As shown in this review, different methods for radiolabeling NPs have been proposed. Therefore, it is essential to standardize these processes and develop reproducible protocols to apply them for clinical purposes.

## Figures and Tables

**Figure 1 biomolecules-13-01241-f001:**
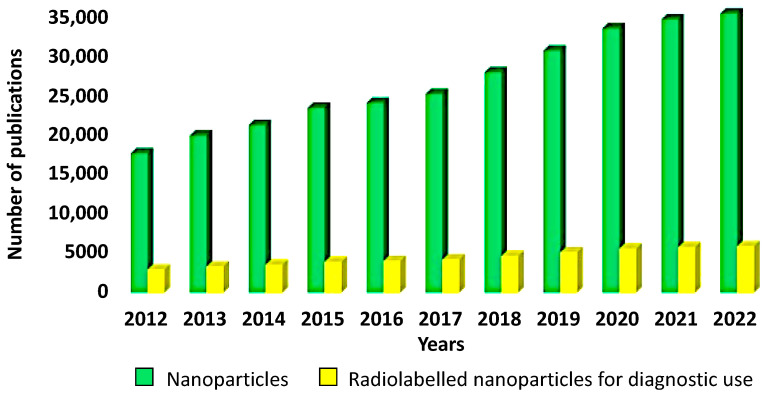
Publications on nanoparticles and on radiolabeled nanoparticles for diagnostic SPECT/PET use from 2012 to 2022.

**Table 1 biomolecules-13-01241-t001:** Particles and labeling methods with Auger’s electron emitter.

Isotope	NP	Type of Radiolabeling	Functionalization	RCY	Applications	Stage of Research	Reference
**^64^Cu**	Copper sulfide NPs	Direct incubation	n.d.a.	>99%	Radiotherapy (RT) and photothermal therapy (PTT) of breast cancer	Tested on human breast cancer xenograft	[6]
	Chitosan NPs	Direct incubation with stirring	n.d.a.	n.d.a.	Cancer imaging and therapy	n.d.a.	[7]
	Melanin NPs	Direct	Coated with PEG	90%	Targeted radionuclide therapy	Tested on epidermoid carcinoma xenograft	[8]
	Feraheme nanoparticle (FH-NPs)	Direct incubation	Coated with carboxymethyl-dextran (CMD)	>85%	Cancer therapy	n.d.a.	[9]
	Boron-nitride nanotubes (BNNTs)	Direct (encapsulation)	n.d.a.	n.d.a.	Cancer diagnosis and therapy	n.d.a.	[10]
	Covellite nanocrystals (CuS NCs)	Direct (encapsulation)	Coated with PEG	49%	Photothermal probes in tumor-ablation treatments	n.d.a.	[11,12]
	Barium-sulphate NPs	Direct incubation	n.d.a.	69%	Targeted alpha therapy (TAT)	n.d.a.	[13]
	Hydroxyapatite/ tenorite NPs	Direct (neutron irradiation)	Folic acid	n.d.a.	Treatment and diagnosis of osteosarcoma	n.d.a.	[14]
	Shell cross-linked NPs (SCKs)	With TETA	Folate	15–20%	Therapy for tumors overexpressing the folate receptor (FR)	Tested on keratin-forming tumor cell line HeLa	[15]

n.d.a. = no data available; NPs = nanoparticles.

**Table 2 biomolecules-13-01241-t002:** Particles and labeling methods with α emitters.

Isotope	NP	Type of Radiolabeling	Functionalization	RCY	Applications	Stage of Research	Reference
**^211^At**	Gold NPs	Direct incubation	Substance P(5–11)	>99%	Radionuclide therapy	Tested on glioblastoma multiforme cells	[19]
			Trastuzumab	>99%	Nano-brachy-therapy for HER2-positive breast cancer	Tested on ovarian adenocarcinoma cells	[20]
			n.d.a.	n.d.a.	Local radiation therapy for cancer	Tested on rat glioma and epithelioid carcinoma of pancreatic duct xenografts	[21]
	Gold nanostars	Direct incubation	n.d.a.	>99%	Targeted alpha-particle therapy (TAT)	Tested on human glioma xenografts	[22]
	Silver NPs	With chloramine-T method	Coated with PEO	>95%	Therapy of small tumors and metastases	n.d.a.	[23]
**^225^Ac**	Superpara-magnetic iron-oxide-based NPs (SPIONs)	Direct incubation	Trastuzumab	>98%	Radioimmunotherapy and magnetic hyperthermia	Study of the biodistribution profiles in SCID mice	[24]
	Liposomes	Direct (encapsulation)	n.d.a.	n.d.a.	Therapy of metastatic cancers	n.d.a.	[25]
	Gold-coated lanthanide phosphate NPs	Direct incubation	Coated with gold	>99%	Targeted alpha therapy (TAT)	n.d.a.	[26]
	LaPO_4_ (monazite) NPs	Direct incubation	Monoclonal antibody 201B	66 ± 4%	Targeted radiotherapy	Study of the biodistribution profiles in healthy BALB/c mice	[27]
	Gold NPs	With TADOTAGA	n.d.a.	86.0 ± 1.8%	Targeted radionuclide therapy	Tested on human glioblastoma xenografts	[28]
	Single-wall carbon nanotube (SWCNT)	With DOTA	Tumor neovascular-targeting antibody E4G10	n.d.a.	Targeted radioimmunotherapy	Tested on murine xenograft model of human colon carcinoma	[29]
			Morpholino oligonucleotide complementary to a modified antibody (cMORF)	n.d.a.	Cancer therapy	Tested on Burkitt’s lymphoma xenografts	[30]
	rHDL NPs	With DOTA	n.d.a.	n.d.a.	Therapy for tumors overexpressing SR-BI proteins	Stuy of the biodistribution profile in healthy mice	[31]
	GdVO4 NPs	Direct (encapsulation)	n.d.a.	67.9 ± 2.4%	Targeted alpha therapy (TAT)	Tested radiochemical yield in vitro	[32]
**^223^Ra**	Liposomes	Direct (encapsulation)	Folic acid (FA) F(ab’)_2_	95 ± 2%	Targeted radiotherapy of cancer	n.d.a.	[33]
	Nanozeolite bioconjugates	Direct incubation	Substance P (5–11) peptide fragment	n.d.a.	Therapeutic construct for targeting glioma cells	n.d.a.	[34]
	Nanomicelles	Direct incubation	n.d.a.	n.d.a.	Targeted radionuclide therapy for bone tumors	Tested on osteosarcoma cell lines	[35]
	Feraheme nanoparticle (FH-NPs)	Direct incubation	Coated with carboxymethyl-dextran (CMD)	n.d.a.	Therapy for cancers	n.d.a.	[36]
	Barium ferrite (BaFe) NPs	Direct incubation	Trastuzumab	61.3 ± 1.8%	Targeted α-therapy	Tested on ovarian adenocarcinoma spheroids	[37]
	Hydroxyapatite and titanium dioxide NPs	Direct incubation	n.d.a.	≥94%	Radionuclide therapy	n.d.a.	[38]
	Folic-acid-functionalized graphene quantum dots (GQD-FA)	Direct incubation	Folic acid	n.d.a.	Antiviral effect against Zika-virus infection	Testing of in vitro antiviral effect against replication on ZIKV infection	[39]
	SPION NPs	Direct incubation	n.d.a.	99%	Targeted alpha therapy (TAT)	Testing of in vitro stability in PBS, bovine plasma and serum	[40]
	GdVO4 NPs	Direct (encapsulation)	n.d.a.	72.3 ± 3.0%	Targeted alpha therapy (TAT)	Testing radiochemical yield in vitro	[32]

n.d.a. = no data available; NPs = nanoparticles.

**Table 3 biomolecules-13-01241-t003:** Particles and labeling methods with β−emitters.

Isotope	NP	Type of Radiolabeling	Functionalization	RCY	Applications	Stage of Research	Reference
**^198^Au**	PAMAM dendrimers	Direct (neutron irradiations)	n.d.a.	n.d.a.	Tumor nano-brachytherapy	Tested on mouse-melanoma-tumor model	[48]
	Gold NPs		Coated with gum arabic glycoprotein (GA)	n.d.a.	Prostate cancer therapy	Tested on mice bearing human-prostate-tumor xenografts	[49]
			Coated with epigallocatechin-gallate (EGCg)	≥99%			[50]
			Coated with mangiferin-a glucose (MGF)	97%	PC-3 prostate-tumor therapy	Tested on mice bearing human-prostate-tumor xenografts	[51]
			n.d.a.	n.d.a.	Bone cancer therapy	Testing of cytotoxicity effect on human osteosarcoma	[35]
			Cyclic arginine−glycine−aspartate peptide (RGD)	>99%	Tumor targeting	Tested on melanoma-tumor-bearing mice	[52]
			Human serum albumin (alb-AuNP) or apolipoprotein E (apoE-AuNP)	n.d.a.	Organ targeting	Study of biodistribution profiles in healthy mice	[53]
			n.d.a.	n.d.a.	n.d.a.	Study of biodistribution profiles in healthy Sprague Dawley rats	[54]
**^90^Y**	Magnetic NPs (MNPs)	Direct incubation	Coated with polyethylene glycol 600 diacid (PEG600)	97% for Fe_3_O_4_-naked and 99% for Fe_3_O_4_-PEG600	Hyperthermia-based cancer treatments	Study of biodistribution profiled in healthy Wistar rats	[55]
	Iron-oxide NPs (IONPs)	Direct incubation	Coated with citric acid, poly(acrylic acid) (PAA) and poly(ethylene glycol)	>99%	Diagnosis and dual magnetic hyperthermia/radionuclide cancer therapy	Tested on murine colorectal carcinoma cells	[56]
	Magnetic NPs (MNPs)	Direct incubation	Coated with imidodiphosphate (IDP) and inositol hexaphosphate (IHP)	>98%	Radionuclide therapy	Study of biodistribution profiles in healthy Wistar rats	[57]
	Feraheme NPs (FH-NPs)	Direct incubation at 120–130 °C	Coated with cCarboxymethyl-dextran (CMD)	>85%	Therapy for cancers	n.d.a.	[9]
	Chitosan NPs	Direct incubation with stirring	n.d.a.	n.d.a.	Cancer imaging and therapy	Study of biodistribution profile in healthy mice	[7]
	YPO_4_:Er^3+^−Yb^3+^ NPs	Direct incubation	n.d.a.	>95%	Optical-based imaging and alternating current (AC) field-based hyperthermia	n.d.a.	[58]
	Lipid NPs	With DTPA	Anti–Flk-1 MAb	>95%	Anti-angiogenesis therapy	Tested on melanoma- and colon-carcinoma-bearing mice	[59]
	HPMA NPs	With DOTA	n.d.a.	n.d.a.	Plasmonic photothermal therapy (PPTT) for prostate tumors	Tested on prostate-tumor-bearing mice	[60]
**^177^Lu**	Iron-oxide NPs (IONPs)	Direct incubation	Coated with citric acid, poly(acrylic acid) (PAA) and poly(ethylene glycol)	>90%	Diagnosis and dual magnetic hyperthermia/radionuclide cancer therapy	Tested on colon carcinoma cell line	[56]
			Coated with alginic acid (MA)	95.21 ± 1.28% for ^177^Lu-MA and 93.65 ± 1.03% for ^177^Lu-MA-PEG	Nano-brachytherapy	Tested on murine mammary carcinoma cell line	[61]
	Feraheme NPs (FH-NPs)	Direct incubation at 120–130 °C	Coated with carboxymethyldextran (CMD)	>85%	n.d.a.	n.d.a.	[9]
	Magnetic NPs (MNPs)	Direct incubation at RT and 80 °C	Poly-L-lysine, proline, and tryptophan	99%	Cancer diagnosis and radionuclide-hyperthermia therapy	n.d.a.	[62]
		Direct incubation	Coated with IHP, IDP, hydroxyethylidene diphosphonic acid (HEDP), and methanediylbis(phosphonic acid) (MDP)	95%	Hyperthermia cancer therapy	n.d.a.	[63]
	Chitosan NPs	Direct incubation with stirring	n.d.a.	n.d.a.	n.d.a.	Evaluated in epithelial lung cancer cells	[7]
	YPO_4_:Er^3+^−Yb^3+^ NPs	Direct incubation	n.d.a.	>95%	Optical-based imaging and alternating current (AC) field-based hyperthermia	n.d.a.	[59]
	Barium-sulfate NPs	Direct incubation	n.d.a.	69%	Targeted alpha therapy of cancer	n.d.a.	[13]
	NaGdF4/Ho−Yb NPs	Direct incubaion	Coated with mesoporous silica (m-SiO2)	98.4 ± 0.4%	Radionuclide therapy	n.d.a.	[64]
	Lu2O3 NPs	Direct (encapsulation)	n.d.a.	>85%	Cancer diagnostics and therapy	Tested on mice bearing melanoma tumors	[65]
	Hydroxyapatite (HA) particles	Direct incubation	n.d.a.	99.1 ± 0.2%	Intra-arterial liver cancer therapy	Study of biodistribution profiles in healthy Wistar rats	[66]
	Fe3O4 NPs	With DP-PEG	Coated with diphosphonate-polyethylene glycol (DP-PEG)	67.4%	SPECT and radiotherapy	Tested on murine breast cancer	[67]
	PAMAM-dendrimer	With DOTA	Lys1Lys3 (DOTA)-bombesin (BN) peptide	97.72 ± 0.26%	Peptide-receptor radionuclide therapy (PRRT)	Tested on infiltrating ductal carcinoma of breast-tumor-bearing mice	[68]
			Lys1 Lys3 (DOTA)-bombesin and folic acid	72%	Targeted radiotherapy	Evaluated in breast tumors over-expressing GRPR and FR cell line	[69]
	HA-PLGA NPs	With DOTA	Hyaluronic acid (HA)	n.d.a.	Radio-synovectomy	Tested on murine macrophage cell line	[70]
	PLGA NPs	With DOTA	Lys1Lys3(DOTA)-bombesin	n.d.a.	GRPr-positive breast cancer therapy	Tested on breast-tumor-bearing mice	[71]
	Dendrimers	With DOTA	n.d.a.	>98%	Cancer therapy	Tested on melanoma-bearing mice	[72]
		Direct incubation	Folate-bombesin	n.d.a.	Plasmonic, photothermal therapy, and targeted radiotherapy of cancers	Tested on breast ductal carcinoma cells	[73]
		With DO3A-py(NO-C)	n.d.a.	n.d.a.	Basis for targeted therapy	Validation of labeling method and biodistribution in rats	[74]
	Micelles	With DOTA	n.d.a.	n.d.a.	Liquid brachytherapy	Tested on murine-colon-carcinoma-bearing mice	[75]
	Nanostar	With DOTA	n.d.a.	>99%	Endo-radiotherapy	Tested on colon cancer isografts and pancreatic cancer xenografts	[76]
	rHDL NPs	With DOTA	Cholesterol	>95%	Targeted radiotherapy	Tested on breast ductal carcinoma xenografts	[77]
	Carbon nanospheres	With DOTA	G3-cNGR peptide	80%	Targeted radio-nanomedicine	Tested on melanoma-bearing mice	[78]
	Fluorescent core-shell silica NPs	With DOTA	n.d.a.	>95%	Targeted radionuclide therapy	Tested on melanoma-bearing mice	[79]
	f-Gd₃N@CC₈₀ nanoplatform	With DOTA	n.d.a.	80%	Brachytherapy	Tested on human glioblastoma xenografts	[80]
	Nanobody	With DOTA	n.d.a.	98%	Radioimmunotherapy for HER2-positive breast cancer	Tested on HER+ ovarian-adenocarcinoma- and melanoma-tumor-bearing mice	[81]
		With DTPA	n.d.a.	98%	Radioimmunotherapy for HER2-positive breast cancer	Tested on HER+ ovarian-adenocarcinoma- and melanoma-tumor-bearing mice	
	Lipid-based NPs	With DTPA	Coated with PEG	>80%	Treatment in oncology	n.d.a.	[82]
	PLGA NPs	With DOTA-TATE	Coated with PEG	98%	Peptide-receptor-radionuclide therapy (PRRT) for NETs	Study of biodistribution profile in healthy Wistar albino rats	[83]
	AuNPs	With DOTA	Panitumumab	n.d.a.	Brachytherapy treatment for locally advanced breast cancer	Tested on metastatic-breast-adenocarcinoma-bearing mice	[84]
			Cyclic Arg-Gly-Asp (RGD) sequence	n.d.a.	Targeted radionuclide therapy for tumors expressing α(V)β(3) integrins.	Tested on human α(V)β(3)-positive glioblastoma xenografts	[85]
			Trastuzumab	n.d.a.	HER2-overexpressing breast cancer therapy	Tested on human breast cancer xenografts overexpressing HER2	[86]
			Substance P (SP) peptides	>95%	Treatment of localized glioblastoma multiforme	Testing of radiobiological effects on human glioblastoma astrocytoma cells	[87]
			Lys3-bombesin	n.d.a.	Radiotherapy and thermal ablation	Tested on PC3-overexpressing cell lines	[88]
			Lys^1^Lys^3^-bombesin (BN) peptide	n.d.a.	Peptide-receptor-radionuclide therapy (PRRT)	Tested on breast-ductal-carcinoma-bearing mice	[73]
			Cyclo(Arg-Gly-Asp-Phe-Lys)Cy ((RGDfK)C)	69%	Targeted radionuclide therapy for tumors expressing α(n)β(3) integrins	Tested on α(V)β(3) integrin-positive glioma tumors in mice	[89]
		Direct incubation at 90 °C	Tyr3-octreotate	>98%	Tumoral-fibrosis therapy	Tested on HeLa human cervical cancer cells	[90]
**^131^I**	Mag-poly(HEMA–MAPA) NPs	With Iodogen	n.d.a.	95–100%	Cancer therapy	Study of biodistribution profiles in healthy Wistar albino rats	[91]
	Magnetic NPs (Fe3O4)	With Iodogen	Thymoquinone glucuronide	95%	Treatment and diagnosis of lung cancer	Tested on human lung cancer xenograft	[92]
	Mesoporous silica NPs	With Iodogen	n.d.a.	95.5 ± 1.2%	Photodynamic therapy (PDT)	Tested on pancreas adenocarcinoma cells	[93]
	PAMAM dendrimer	With Iodogen	Vascular targeting peptide (VTP)	77.6%	Theranostic nano-sensors for medullary thyroid carcinoma	Tested on medullary carcinoma cells	[94]
		With chloramine-T method	3-(4′-hydroxyphenyl)-propionic acid-OSu (HPAO) and folic acid (FA)	97.7 ± 0.7%	Imaging and radiotherapy for tumors	Tested on rat glioma bearing mice	[95]
		With chloramine-T method	LyP-1 peptide	61.95 ± 3.18%	SPECT imaging, radionuclide therapy, and metastasis therapy for cancer	Tested on stage IV human breast cancer xenograft	[96]
	Polyethylenimine-entrapped gold NPs (Au PENPs)	With HPAO moiety	Chlorotoxin	60.4 ± 5.4%	Imaging and radionuclide therapy for glioma	Tested on rat-glioma-bearing mice	[97]
			Buthus martensii Karsch chlorotoxin (BmK CT)	77.0 ± 4.97%	Imaging and radionuclide therapy for glioma	Tested on rat-glioma-bearing mice	[98]
	Single-walled carbon nanotubes (SWNTs)	With chloramine-T method	Coated with polydopamine (PDA)	~90%	Multimodal-imaging-guided combination therapy for cancer	Tested on stage IV human breast cancer xenografts	[99]
	Albumin nanospheres	With SPDP	anti-AFPMcAb	65%	Radio-chemotherapy of hepatoma	Tested on human hepatoma xenografts	[100]
	Albumin-paclitaxel NPs	With chloramine-T method	Paclitaxel (PTX)	n.d.a.	Combined chemotherapy and radioisotope therapy (RIT) of cancer	Tested on stage IV human-breast-cancer xenografts	[101]
	BSA–PCL NPs	With chloramine-T method	Anti-epidermal growth factor receptor (EGFR–BSA–PCL)	50–85%	Radionuclide therapy in EGFR-overexpressing tumors	Tested on stage IV lung-adenocarcinoma xenografts	[102]
	BSA@CuS NPs	With chloramine-T method	n.d.a.	66–80%	Radiotherapy and photo-thermal-therapy of anaplastic thyroid carcinoma (ATC)	Tested on thyroid-gland-anaplastic-carcinoma xenografts	[103]
	Reduced nano-graphene oxide (RGO)	With chloramine-T method	Polyethylene glycol (PEG) coated	80%	Combined chemotherapy and radionuclide therapy (RIT) for cancer	Tested on stage IV human-breast-cancer xenografts	[104]
	Transferrin-capped polypyrrole NPs	With chloramine-T method	Transferrin	90%	Photothermal radiotherapy	Tested on glioblastoma xenograft	[105]
	Albumin-encapsulated liposomes	With chloramine-T method (encapsulation)	n.d.a.	90%	Internal radioisotope therapy (RIT)	Tested on stage IV human-breast-cancer xenografts	[106]
	Albumin-based gadolinium oxide NPs (GdNPs)	With chloramine-T method	Bone-seeking alendronate	>95%	Therapeutic monitoring of primary bone tumors and metastases	Tested on rat model of focal malignant osteolysis	[107]
	Polyethylene-glycol-capped silver NPs	Direct (encapsulation)	Coated with PEG	98.0 ± 0.8%	Tumor theranostic probe	Tested on solid-tumor-sarcoma-bearing mice	[108]
	Hollow copper-sulfide NPs and paclitaxel	Direct with NaClO_4_	Paclitaxel	95.8 ± 1.3%.	Theranostic agent for orthotopic breast cancer	Tested on orthotopic breast cancer xenografts	[109]
	Magnetic NPs (MNPs)	Carbodiimide method with EDC	Poly-l-lysine, proline, and tryptophan	99%	Cancer diagnosis and radionuclide-hyperthermia therapy	Study of biodistribution profiles in healthy Wistar rats	[71]
	Ultrasmall Pd nanosheets	Direct incubation with stirring	Coated with PEG	>98%	Photoacoustic imaging and combined photothermal and radiotherapy	Tested on stage IV human breast cancer xenograft, on spontaneous mammary tumor in BALB/c mouse model, and on a Mst1/2 double-knockout hepatoma model	[110]
**^166^Ho**	Holmium acetylacetonate NPs	Direct (neutron irradiations)	n.d.a.	n.d.a.	Intra-tumoral radionuclide treatment for solid malignancies	Tested on rabbit anaplastic squamous cell carcinoma	[111]
	Holmium iron garnet NPs	Direct (neutron irradiations)	n.d.a.	n.d.a.	Treatment of skin cancer	n.d.a.	[112]
	Iron-oxide NPs	Direct incubation with stirring	Vascular targeting peptide (VTP)	>95%	Treatment of knee arthritis	n.d.a.	[113]
	Chitosan NPs	Direct incubation with stirring	n.d.a.	n.d.a.	Cancer imaging and therapy	n.d.a.	[7]

n.d.a. = no data available; NPs = nanoparticles.

## Data Availability

All papers and data are available upon request to M.V.

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
