# Peer review of "Methods for Radiolabeling Nanoparticles (Part 3): Therapeutic Use"

_biomolecules, 2023, doi:10.3390/biom13081241_

Round 1

Author Response

Reply enclosed

Reviewer 2 Report

Introduction, page 1, lines 26 and 29 authors refer to figures that does not exist in the text. Please remove or correct.

Page 2, chapter 2. Copper-64 is not typical representative of Auger emitters like e.g.: 111In, 67Ga, 99mTc, 195mPt, 125I and 123I. I would like to suggest yet another Auger-emitter, the Tb-161 that is currently of high interest.

Page 3, line 126, please correct the reference 12 to Xie et al.

Page 4, line 158 - Please correct. I disagree with a statement that till today nobody used e.g. DOTA to label NPs with Cu-64.  Please re-check the literature available. In some previous work several chelators were tested for Cu isotopes to label polymers (see https://doi.org/10.1524/ract.2009.1669),  DOTA was used to label magnetic NPs with Cu-64 (see: https://doi.org/10.1021/bc900511j) , for a review of other Cu-64 labelled NPs see https://doi.org/10.1089/cbr.2009.0674).

Page 5, table 2, I would suggest adding 223Ra labelled SPIONs, see e.g.: https://doi.org/10.1515/ract-2019-3206https://doi.org/10.1007/s11051-016-3615-7 

Page 9, line 354, To make the reference 36 at least partially comprehensive, I suggest adding a reference to the study of recoils release from surface labelled 223Ra-TiO2 NPs of the same research group. See: https://doi.org/10.3390/ma16010343.

Finally, on Page 15, line 623, I suggest to link I-131 chapter with At-211 labelling, since some methods are very similar and also the radiochemistry of iodine and astatine have many common properties.

Author Response

Reply enclosed
